# A Fast Lithium-Ion Battery Impedance and SOC Estimation Method Based on Two-Stage PI Observer

**Tao Chen, Mengmeng Huo, Xiaolong Yang * and Rui Wen**

State Key Laboratory of Advanced Design and Manufacture for Vehicle Body, College of Mechanical and Vehicle Engineering, Hunan University, Changsha 410082, China; chentao97@hnu.edu.cn (T.C.); huomengmeng201209@163.com (M.H.); ruiwenww@126.com (R.W.)

\* Correspondence: xyangusc@163.com

**Abstract:** Due to the complex changes in battery state, the accurate and fast estimation of battery state of charge (SOC) is still a great challenge. Here, a fast estimation method of battery impedance and SOC based on a multi-level PI observer is proposed. The observer model reflects the change of the battery state characteristics through the dynamic impedance, and then the system compensation factor is added to the observer to dynamically adjust the parameters of the battery model. The effectiveness of the algorithm is verified by the compound dynamic stress test (DST) experiment. The results show that the introduction of the compensation factor enables the system to tolerate a certain degree of impedance fluctuation and capacity attenuation and the maximum SOC estimation error can be kept within 2%.

**Keywords:** battery; state of charge; impedance; state observer; compensation factor

## 1. Introduction

To solve the problems of energy crisis and environmental pollution, electric vehicles have achieved more and more attention. Power batteries have developed rapidly as the main energy storage devices for electric vehicles [1]. Accurate and reliable state estimation is one of the core functions of battery management [2]. Battery states usually include state of charge (SOC), state of health (SOH), state of power (SOP), state of energy (SOE) [3–6], where SOC is the most basic and important state estimation parameter. The Battery Management System (BMS) requires a reliable SOC estimation method to determine the current state of the battery, from which the current actual power consumption, battery safety, and battery life are inferred. Different from battery parameters measured directly online, such as voltage and current, SOC cannot be directly obtained by measurement in practical applications [7]. Typically, SOC can be characterized by the amount of active material in the battery, which is affected by many factors, such as voltage, current, temperature, aging, etc. These factors are coupled to each other, which add a lot of difficulty to SOC estimation.

In order to solve the above problems, many SOC estimation methods were proposed [8,9]. Among them, the coulomb counting method is a widely used method for SOC estimation because of its simplicity and effectiveness [10]. However, there are also some problems that cannot be ignored, such as: (1) The signal drift of the sensor and the noise generated by the actual operation of the vehicle lead to the formation of cumulative errors, which greatly reduces the estimation accuracy. (2) Initial value $SOC_0$ is difficult to determine. (3) The actual available capacity of the battery will change dynamically due to factors such as aging, discharge rate, temperature and so on. Many other methods were proposed to solve the drawbacks of the Coulomb counting method. One of the effective methods to improve the accuracy of SOC estimation is to use the unique correlation between the measurable parameters of battery and SOC, such as open-circuit voltage method [11] or electrochemical impedance spectroscopy [12]. However, these methods require long periods of resting for batteries, which is not suitable for practical control and these measurable

parameters are also affected by internal or external factors, such as battery aging [13]. The model-based method is another popular SOC estimation method. The purpose of the model-based method [14] is to gradually converge the output parameters (such as voltage) of the model to the target value during the recursive process and then obtain an estimated value of the battery state parameters. The model-based method requires an accurate battery model to correctly reflect the battery characteristics and they estimate battery SOC using typical regression algorithms of modern control theory, including least squares [15], filters [16,17], neural networks [18], fuzzy control algorithm [19], sliding mode observer [20] and other methods. Meanwhile, charge imbalance is a very common problem in multi-battery SOC estimation. The module-based battery charge balancing system [21] proposes a mathematical model to accelerate the battery charge balancing and improve the system performance.

The model-based method has higher computational accuracy and can solve the problem of the uncertain initial value in SOC estimation. The Kalman-based method is one of the popular model-based models. It can get high accurate estimations based on known noise statistics even when the initial SOC is unknown. However, this method still requires an accurate battery model and is applicable only in some specific cases. Neural networks can be adapted to all batteries, but need a lot of data for training. Sliding mode observers can suppress external disturbances and modeling errors, but their biggest drawback is the chattering effects. In addition, rapid changes of battery parameters in actual conditions may cause the estimation method to diverge due to its inability to adapt. The adaptive algorithm can evaluate and correct unknown or uncertain system model parameters and noise statistical characteristics. However, its stability is relatively low and the convergence analysis method lacks universality.

To improve the accuracy and reliability of the SOC estimation for the real engineering problem, a two-stage PI observer method is used to estimate the impedance and SOC of lithium-ion batteries. The integrator of the PI observer can give the observer better robustness in modeling uncertainty. It can improve the accuracy by suppressing the system interference and speed up the computation efficiency of the SOC estimation [22]. It is well known that the capacity and impedance change with battery aging, which further affects the estimation results of SOC. Thus, the online calculated dynamic impedance can be used to initially determine the status of battery usage. A compensation factor ξ is added to modify the impedance of the observer model, which is used to account for the effect of battery aging. It is derived from the average dynamic impedance during the last discharge cycle of batteries and used to compensate for part of the system error due to the attenuation capacity during the current discharge cycle. Thus, the method can automatically match the PI observer at different battery capacities or other conditions by the influence of the factor above, instead of modifying the parameters of the PI observer itself. Comparing with the method based on several observers such as a sliding mode observer [20] and a dual-circuit observer [23], the PI observer is fast and accurate. Although the sliding mode observer is easy to design, it may encounter the problems of large chatting effect and poor tracking performance, while the double-loop observer has lower complexity but lower accuracy. The proposed observer can improve the estimation performance in different usage of the battery while ensuring a simple calculation.

This paper is organized as follows: Section 2 introduces the SOC estimation method based on multi-level PI state observer. In Section 3, the proposed method will be validated under the conditions of the composite dynamic stress test (DST) and the results will be comprehensively analyzed. Finally, the conclusions of this study will be given in Section 4.

## 2. SOC Estimation Based on PI Observers

The accuracy of battery measured parameters and battery aging state are considered as the main factors affecting SOC estimation here. The dynamic impedance technique and two PI state observers will be combined into an estimation method for SOC and battery impedance. The overall structure of the two-stage PI observer system is shown in

Figure 1. The left in Figure 1 is the first-level PI observer which is used for estimating the battery impedance, and the right is the second-level PI observer which is used for the SOC estimation. Next, a compensation factor ξ is also introduced in the system to compensate for part of the capacity loss due to usage of battery, which is obtained by the average of battery impedance at last cycle.

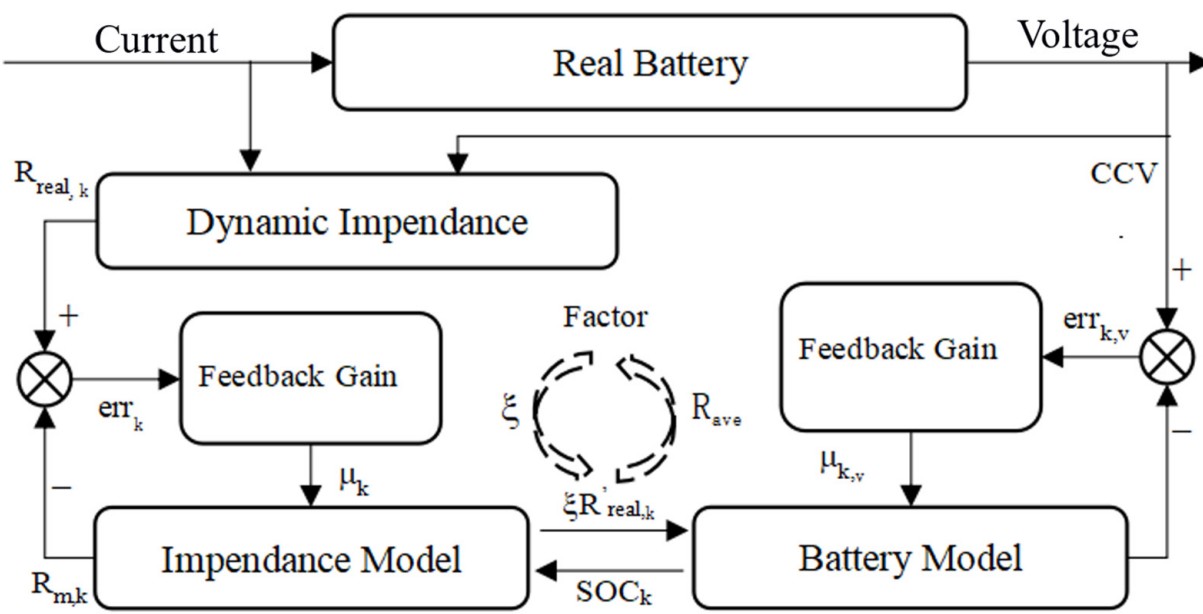

**Figure 1.** Multi-level PI observer block diagram for impedance and SOC.

### 2.1. Introduction of the Electrochemical Battery Model

For the two stages of PI observer, a battery model is needed first. Compared with the complex battery models such as the distributed parameter equivalent circuit model [24] and the distributed electrochemical model [25], a combined model consists of Shepherd model, Nernst model, and Unnewehr universal model [26] is more suitable for the PI observer. This combined model is more able to directly reflect the external characteristic relationship between voltage and SOC of battery, which is formulated as follows:

$$CCV = E_0 - R{\cdot}I - k_0{\cdot}soc - \frac{k_1}{soc} + k_2{\cdot}\ln(soc) + k_3{\cdot}\ln(1 - soc) \tag{1}$$

where *CCV* is battery terminal voltage. *I* represents measured current, which is positive during discharge and negative for charging. *R* is battery impedance. $E_0, k_0, k_1, k_2, k_3$ is the undetermined coefficient of the battery model. Note the battery impedance *R* will vary at different battery SOC and usage.

According to the above battery model structure in Equation (1), the change in the battery terminal voltage can be considered mainly to be the result of the SOC change and the voltage division caused by the battery impedance *R*. In general, inaccurate SOC, parameter measurement errors, and changes in battery impedance can also cause errors in the output of the above model.

### 2.2. First-Level PI Observer for Impedance Estimation

Considering the possible influence of system noise and measurement noise in practical application, a PI observer will be used to obtain the battery dynamic impedance to eliminate the influence of noise. A simple and fast calculation of dynamic impedance [27] can be defined as the change rate of voltage with the change of current. It can be expressed as:

$$R_{\zeta,k} = \frac{\Delta U}{\Delta I} = \frac{|U_k - U_{k-1}|}{|I_k - I_{k-1}|} \tag{2}$$

where $R_{\zeta,k}$ is the battery dynamic impedance. $U_k$, $U_{k-1}$, $I_k$, $I_{k-1}$ are the battery measured voltages and current at k and k − 1, respectively. Figure 2 is the result of battery impedance obtained from the relevant experiments at different sampling intervals, taking SOC = 0.5. As shown in Figure 2, when the time interval is between 0 and 0.3 s, the impedance changes gently, but after 0.3 s, the impedance changes sharply. The result declares that the short sampling interval can reduce the influence of current on the calculation result of dynamic impedance. The sampling interval adopted in this paper is 0.1 s.

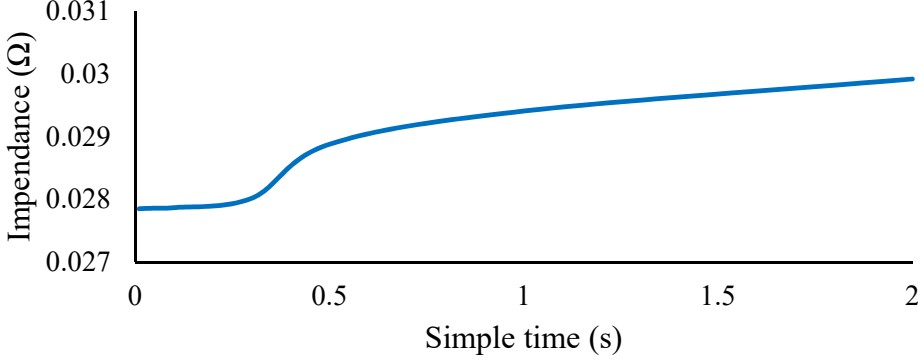

**Figure 2.** Battery impedance at different sampling intervals (SOC = 0.5).

In order to find out the relationship between the dynamic impedance and battery operating current, a set of experiments with different currents at discharge is designed, as shown in Table 1, and the details of the impedance test are shown in Section 3.1. The battery changes from one discharging current to another current. The voltage and current are recorded before and after the transition. Then the impedance is calculated using Equation (2). There are four kinds of current considered, namely I1 to I4. The discharge rate of the current is 1C, 2C, 3C, and 4C, respectively (i.e., $I_1$ is 1C, $I_2$ is 2C, $I_3$ is 3C and $I_4$ is 4C). Seven test cases are carried out and can be divided into three groups, which are able to display the relationship between different dynamic impedance and static impedance. In Table 1, the group 1–3 represent the same initial current, same of the end current, and static impedance test which sets the end current to zero, respectively.

**Table 1.** Battery discharge condition.

| Group | Discharge Condition | |
|---|---|---|
| | $\Delta I_1$ | $I_4 - I_3$ |
| 1 | $\Delta I_2$ | $I_4 - I_2$ |
| | $\Delta I_3$ | $I_4 - I_1$ |
| 2 | $\Delta I_4$ | $I_3 - I_1$ |
| | $\Delta I_5$ | $I_2 - I_1$ |
| 3 | $\Delta I_6$ | $I_4 - 0$ |
| | $\Delta I_7$ | $I_1 - 0$ |

The impedance comparison results are shown in Figure 3. It can be seen that the values of dynamic impedance $R_{\zeta,k}$ and static impedance $R_{real,k}$ are very similar at the same SOC but at different currents. Therefore, it can be considered that the effect of current on impedance can be ignored, that is, $R_{real,k} \approx R_{\zeta,k}$. The small difference in impedance during the discharge period may be caused by the inaccurate measurement parameters during operation.

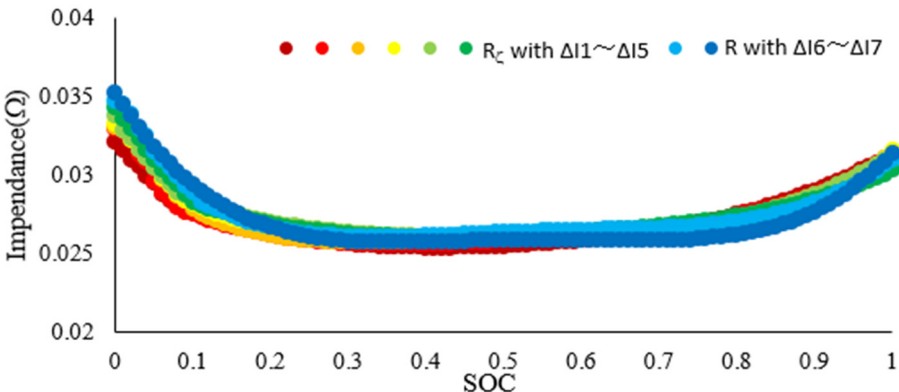

**Figure 3.** Dynamic impedance ($R_\zeta$) and static impedance ($R_{real,k}$) in the case of different current changes.

The PI state observer will be used to obtain the battery impedance that filters out the effects of noise. The basic structure is shown in the feedback loop on the left side of Figure 1. The first-stage PI observer adjusts the feedback gain by scaling the scale factor as follows:

$$\mu_k = K_p \cdot \left( err_k + \frac{1}{T_i} \cdot \sum_{i=0}^{k} err_i \right) \tag{3}$$

where $\mu_k$ is the feedback gain. $K_p$ is the scaling factor that adjusts the feedback gain. $T_i$ represents the time of integration. $err_k$ means the error between the model output and the measured parameters at time k, as shown in Equation (4).

$$err_k = R_{real,k} - R_{m,k} \tag{4}$$

$R_{real,k}$ is the static impedance at time k, as the input signal of the first-level state observer. $R_{m,k}$ represents the output value of the impedance model at time k. This model reflects the relationship between the battery impedance and the SOC, such as Equation (5), expressing the characteristics of the battery impedance variation. Where $\frac{\partial R}{\partial soc}$ is a polynomial about the SOC obtained from experimental data.

$$R_{m,k} = R'_{real,k-1} + \left. \frac{\partial R}{\partial soc} \right|_{soc_{k-1}} \cdot (soc_k - soc_{k-1}) \tag{5}$$

Finally, the output impedance parameter $R'_{real,k}$ based on the PI state observer is as follows:

$$R'_{real,k} = R'_{real,k-1} + \mu_k \tag{6}$$

*2.3. Second-Level PI Observer for SOC Estimation*

The model required for the second-stage PI observer used for SOC estimation at time k, as shown in Equation (7), is a discrete form of Equation (1). The second-stage observer updates its model state parameters according to the output of the first-stage PI state observer, with the battery current and SOC as model inputs, and the battery voltage as the model output.

$$CCV_{m,k} = E_0 - R \cdot I_{t,k} - k_0 \cdot soc_k - \frac{k_1}{soc_k} + k_2 \cdot \ln(soc_k) + k_3 \cdot \ln(1 - soc_k) \tag{7}$$

where $CCV_{m,k}$, $I_{t,k}$ is battery terminal voltage and measured current at k time. The related process of the second-stage PI state observer is shown on the right side of Figure 1, and the feedback adjustment algorithm is as shown in Equation (8):

$$\mu_{k,v} = K_{p,v} \cdot \left( err_{k,v} + \frac{1}{T_{i,v}} \cdot \sum_{i=0}^{k} err_{i,v} \right) \tag{8}$$

where $K_{p,v}$ is the scale factor used to adjust the feedback gain of the observer. $T_{i,v}$ is the time of integration. $err_{i,v}$ is the error between the measured voltage $CCV_k$ and the model voltage $CCV_{m,k}$ at time k, expressed as:

$$err_{k,v} = CCV_k - CCV_{m,k} \tag{9}$$

Therefore, the SOC estimation of the PI observer based on multi-level feedback is derived as follows:

$$SOC_k = SOC_{k-1} + \frac{\eta I_t \cdot T_s}{3600 \cdot C_n} + \mu_{k,v} \tag{10}$$

However, the battery impedance and battery capacity will change with the using of the battery, which will reduce the processing effect of the current observer and the estimation accuracy of the SOC. Therefore, in addition to using the first-stage observer to calculate the battery impedance online, a compensation factor ξ is also introduced in the second-stage state observer model. The compensation factor corrects the impedance parameter in the model to compensate for the partial battery capacity that is attenuated due to operation. The system can realize accurate SOC estimation under different aging degrees of the battery due to the influence of the compensation factor. Therefore, the battery impedance in this model is redefined as shown in Equation (11).

$$R = \xi \cdot R'_{real} \tag{11}$$

where $R'_{real}$ is output signal of the first-stage PI observer, battery impedance. $R$ is the impedance in the model that can adapt to the current second-stage observer.

During the entire cycle of the battery, the observer parameter is treated as a fixed value, and the compensation factor ξ alters the model characteristic parameters according to the usage of the battery. Since the battery SOH usually presents an exponential change with the number of cycles [28], the changes in the compensation factor mentioned will be updated in the form of Equation (12). $R_{fresh}$ is used as the initial value of battery impedance. $R_{ave}$ is the average impedance of the battery obtained under the n[th] battery discharge cycle, to update the observer compensation factor under the (n+1)[th] battery discharge cycle.

$$\xi = \log\left( a * \frac{R_{fresh}}{R_{ave}} + b \right) \tag{12}$$

The method is simple in calculation and high in precision. The compensation factor ξ, can adjust the matching degree between the battery model and the observer according to the battery impedance, and improve the adaptability of the SOC estimation method.

## 3. Verification and Discussion

In this section, the improved experiment based on the Dynamic Stress Test is introduced. The basic parameters of the battery used are listed in Table 2. Next, it will be verified that the proposed PI observer with the compensation factor ξ is universal in different usage of batteries. Finally, the performance of fault tolerance of the compensation factor is analyzed.

**Table 2.** Battery parameters.

| | |
|---|---|
| Battery type | ISR18650PC |
| Battery capacity | 2.6 Ah |
| Working voltage | 4.2–2.75 V |
| Maximum continuous discharge current | 15 A |
| Initial Impedance | $\leq 30.0$ mΩ |

## 3.1. Experiment Design

The experimental data used in this paper are obtained on an experimental platform with high-precision sensors. As shown in Figure 4a, the battery test bench consists of a battery test instrument with eight independent channels, a thermal chamber, and a hosting computer. Figure 4b illustrates the battery test projects, including the characterization test and aging test. Their loading profiles are displayed in Figure 5.

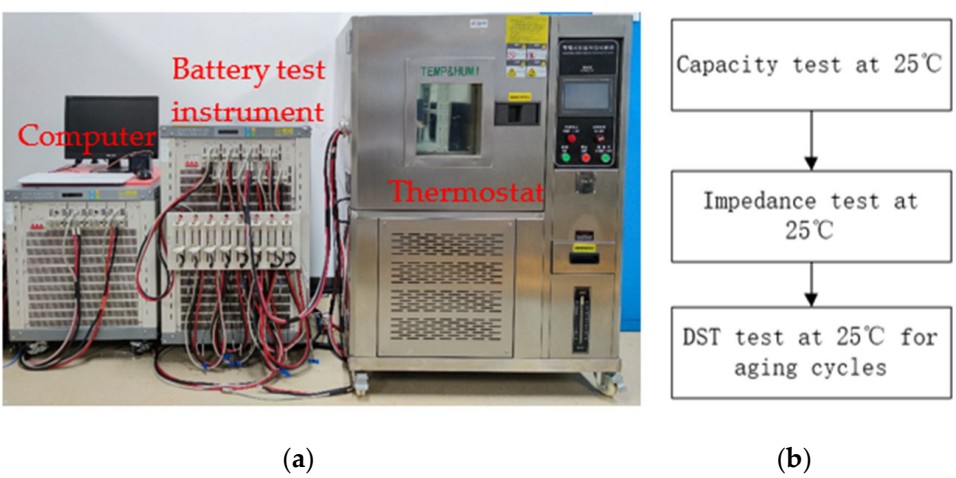

(a)　　　　　　　　　　　　　　　　　　　(b)

**Figure 4.** Battery experiments: (**a**) test bench, (**b**) test schemes.

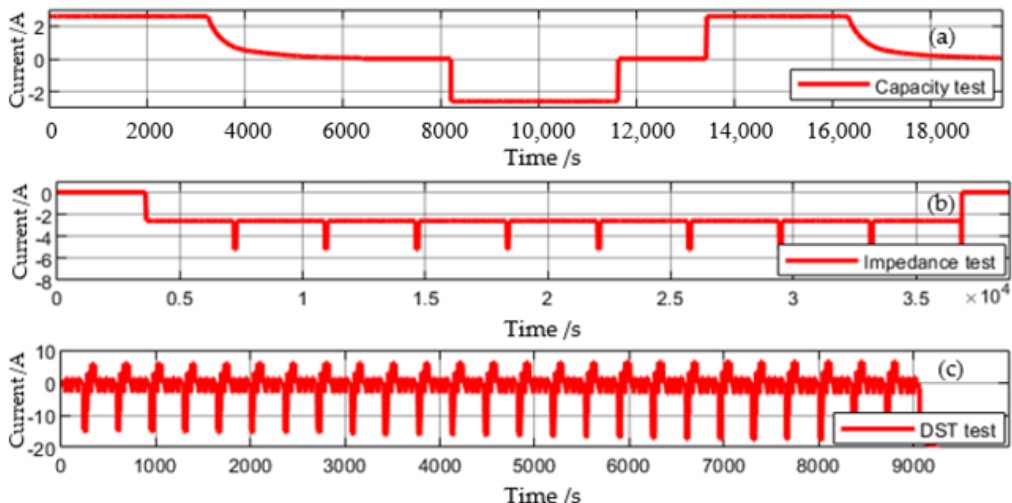

**Figure 5.** Current profiles: (**a**) Capacity test, (**b**) Impedance test, (**c**) DST test.

As shown in Figure 5a, the capacity test is usually executed to measure the battery capacity. The battery is firstly charged at the CC (constant current)-CV (constant voltage) mode and then discharged with a constant current of 1 C until the terminal voltage reaches the discharge cut-off voltage. Figure 5b shows the impedance test at different SOC. The battery is discharged at a current of $I_1$ (e.g., 1C), followed by a current of $I_2$ (e.g., 2C) for 40 s. In this case, the battery releases 10% of its capacity and it executes 10 cycles. Finally, the experimental data are used to calculate the impedance of different SOC according to Equation (2), and the SOC and impedance are fitted by a polynomial as shown in Figure 3. The DST cycles are employed to verify the proposed states estimation method. Its current profile is shown in Figure 5c. A set of battery aging tests are also carried out by the DST cycles.

The battery model described in Equation (2) uses the least−squares method for parameter identification. The results are shown in Table 3. The reference SOC is also

calculated based on the experimental data using the Coulomb counting method. In order to be able to verify the performance of the proposed algorithm, the simulation also simulates the effects of actual conditions including sensing noise, drift current, and unknown capacity.

**Table 3.** Battery model parameters.

| Parameter | Value |
|---|---|
| $E_0$ | 3.459 |
| $k_0$ | $-0.039$ |
| $k_1$ | 0.001 |
| $k_2$ | 0.066 |
| $k_3$ | $-0.070$ |

Based on the Dynamic Stress Test (DST) in the USABC Electric Vehicle Battery Test Manual, the experiment will be modified to simulate the load changes of pure electric vehicles due to road conditions and traffic situations. As shown in Figure 6, based on the characteristics of the driving conditions of pure electric vehicles, three different modifications are made based on the DST standard test. A complete discharge cycle of the battery will be accomplished alternately by four different discharge standards in the figure until the energy is exhausted. The battery voltage, current and discharge energy will be recorded during the experiment. A number of batteries with different degrees of aging are used to verify the accuracy of the proposed estimation method. The current and voltage of the battery under DST cycles as shown in Figure 7.

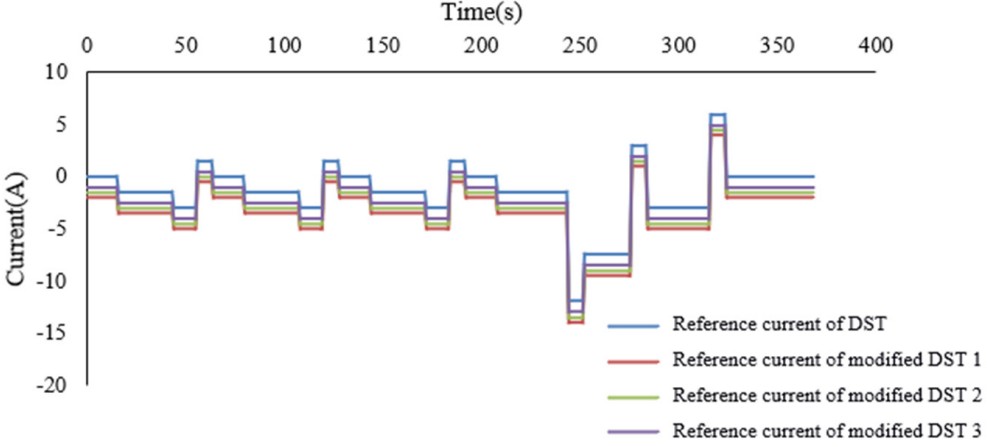

**Figure 6.** Dynamic stress test standard and modified dynamic stress test for a pure electric vehicle.

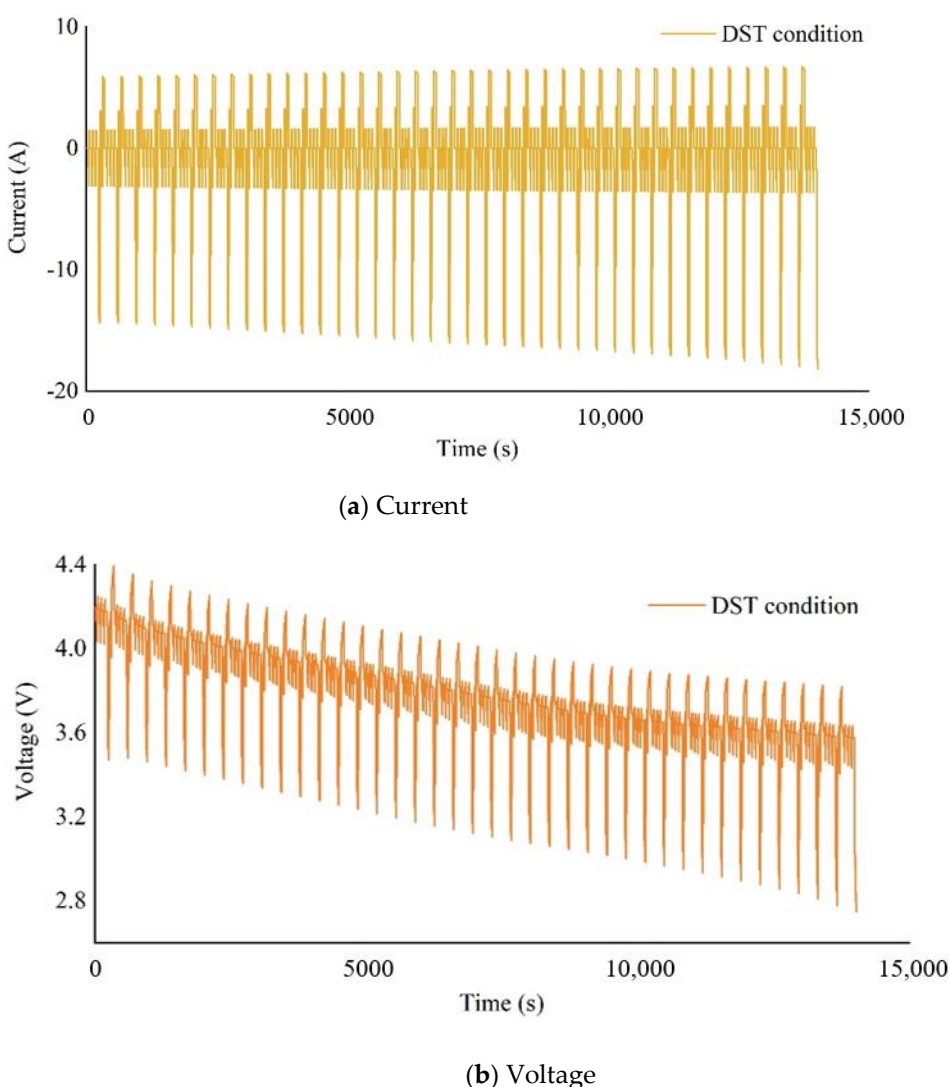

(**a**) Current

(**b**) Voltage

**Figure 7.** Current and voltage under DST cycles (**a**) Current (**b**) Voltage.

### 3.2. Verification and the Results

The new battery does not need to consider the issue of dynamic capacity, so the compensation factor at this time is set to 1. Figure 8 shows the dynamic impedance of a new battery in a complete DST cycle and the battery impedance after processing by the state observer. The dynamic impedance in the Figure is extremely susceptible to measurement noise. Through the first-stage PI state observer, the battery impedance can filter out the effects of noise and show a steady trend of change, which can avoid unreasonable SOC estimates for the second-stage observer. When artificially adding measurement noises or erroneous SOC initial values to the system input signal, the comparison between the SOC in the state observer and the SOC reference in the experiments is shown in Figure 9a. In order to verify the applicability of the proposed SOC estimation method based on the PI observer, the method described in Ref. [29] (DEKF) is also used in the SOC estimation. Figure 9b shows the error between the estimated results and the reference SOC. The errors estimated by the PI observer and the DEKF can be controlled within ±2%. The errors are mainly due to imperfections of the battery model. Meanwhile, compared with the DEKF method, the SOC estimated by the PI observer converges to the exact value more quickly under the wrong initial SOC. The results show that the proposed method can quickly and accurately compensate the initial error of SOC and the cumulative error caused by measurement noise in new batteries.

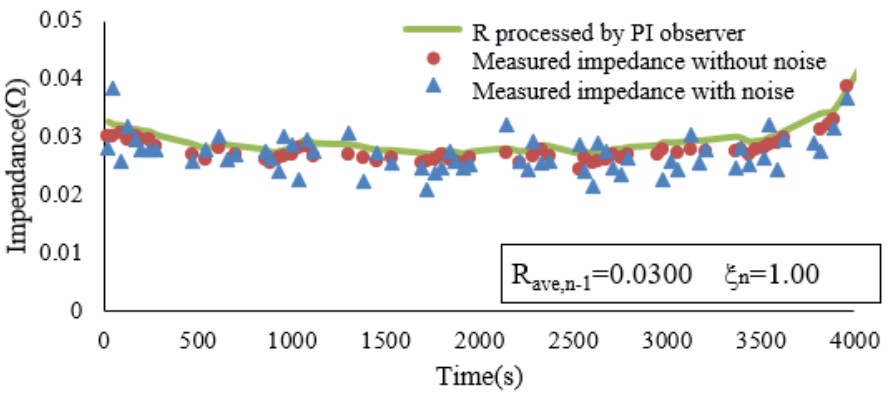

**Figure 8.** Comparison of battery dynamic impedance.

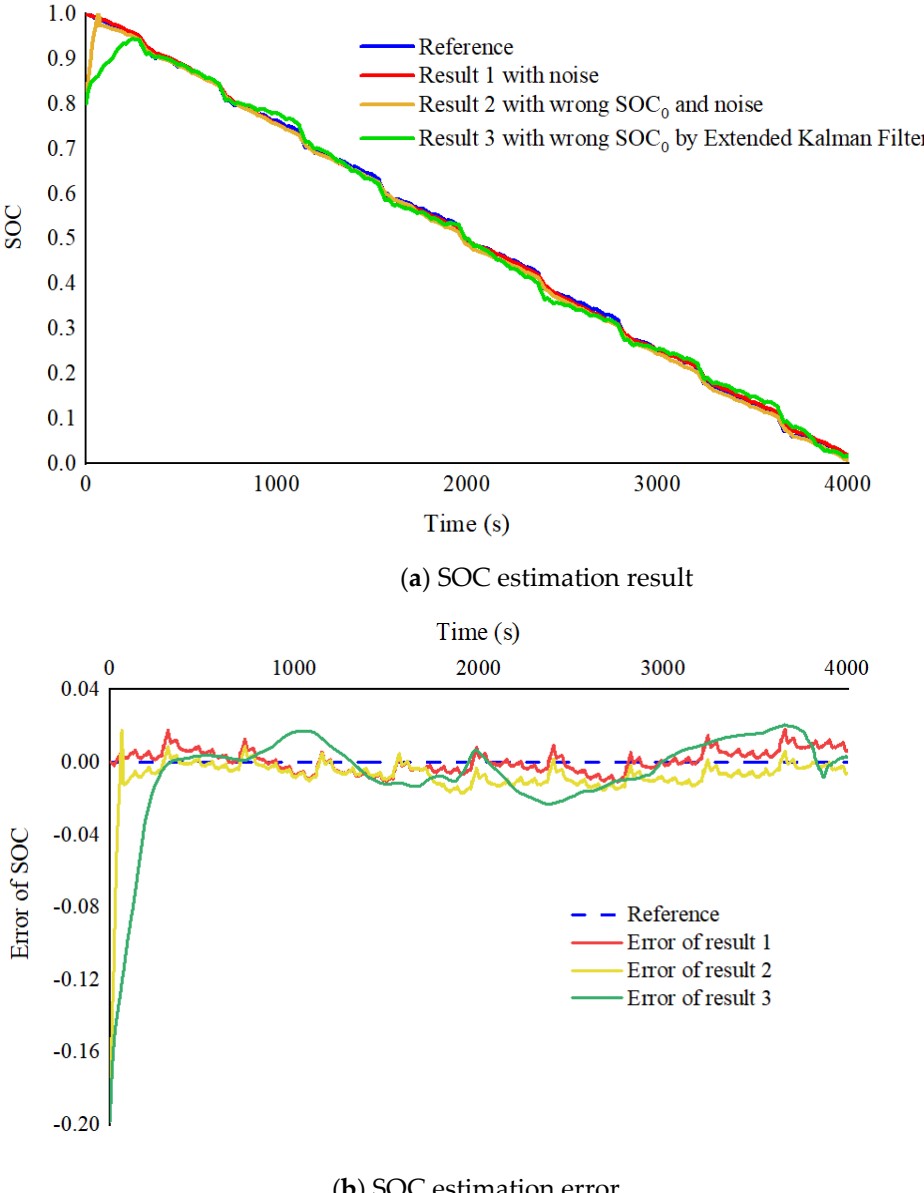

(**a**) SOC estimation result

(**b**) SOC estimation error

**Figure 9.** (**a**) SOC estimation result. (**b**) SOC estimation error.

The compensation factor $\xi$ mentioned is predicted based on the average impedance of the battery obtained under each uninterrupted discharge cycle of the battery. This paper selects the discharge cycle data under different states of a battery to determine the compensation factor. The exact parameter in Equation (12) is obtained by fitting as follows:

$$\begin{cases} \xi = \log\left(a * \frac{R_{fresh}}{R_{ave}} + b\right) \\ including : R_{fresh} = 0.03\ \Omega, a = -1.502, b = 4.216 \end{cases} \tag{13}$$

The battery multiplex DST experimental data under two different usage (after 100 discharges and 300 discharges) was used to verify the SOC estimation method mentioned. The values of the model compensation factor are calculated by Equation (13) to be 1.0999 and 1.211, respectively. Figure 10a,b contains the dynamic impedance $R_{\zeta,k}$ of the battery measured, and model parameter $R$ updated of the second-stage observer. The impedance $R$ is greater than the dynamic impedance $R_{\zeta,k}$ in order to compensate for part of the capacity degradation caused by battery aging. The deeper the battery ages, the greater the deviation between $R$ and $R_{\zeta,k}$.

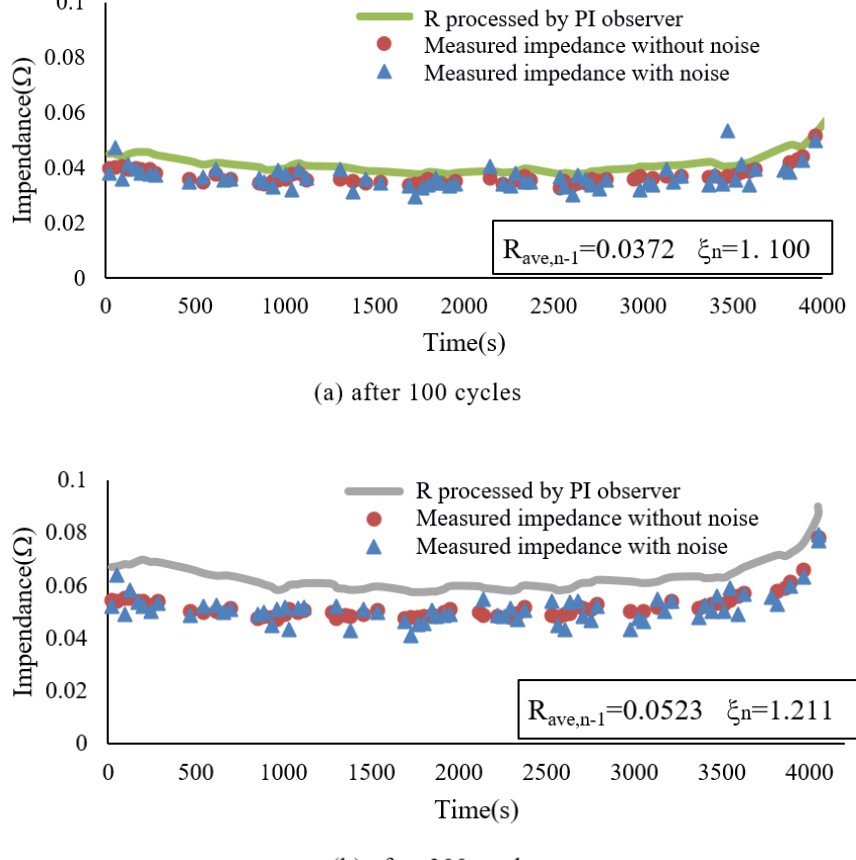

(a) after 100 cycles

(b) after 300 cycles

**Figure 10.** Dynamic impedance of battery and output impedance of observer.

The results of the SOC estimation are shown in Figure 11. The SOC estimation result without compensation factor causes the observer to fail due to the change in capacity and impedance, and the final error >±2%. The SOC estimation result with the compensation value controls the error within ±2%.

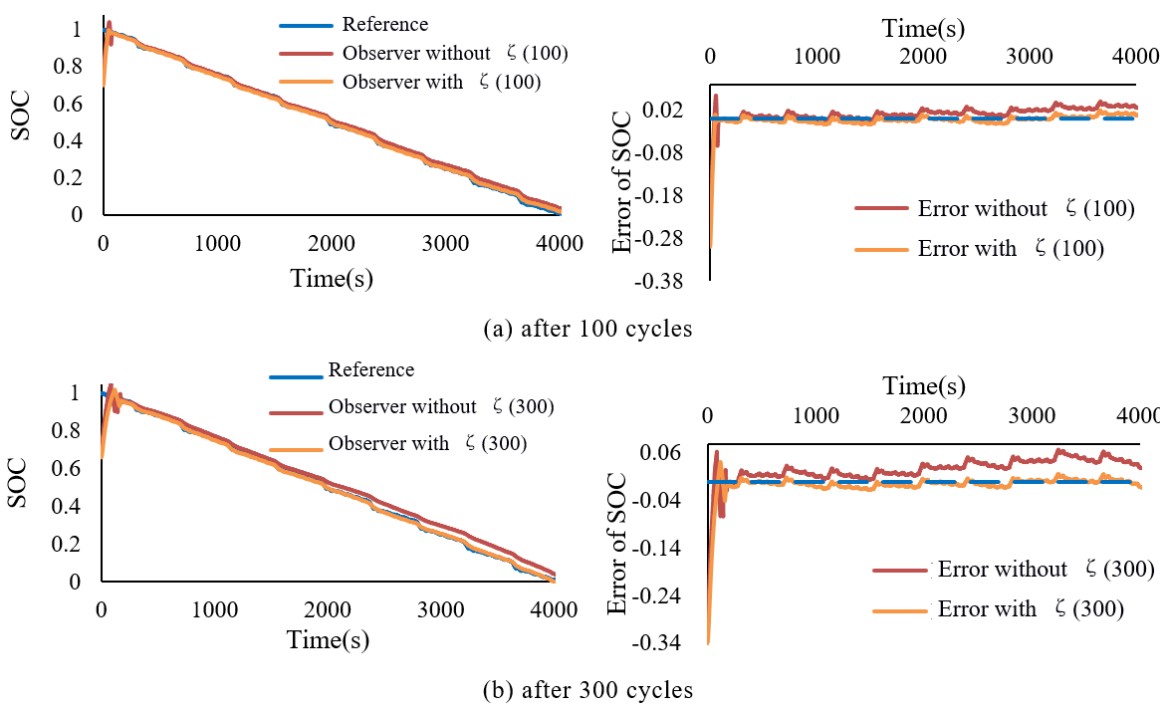

**Figure 11.** Battery SOC estimation results and errors (including with ξ and without ξ).

### 3.3. Analysis for Fault Tolerance of the Factor ξ

Inaccurate results of battery impedance directly affect the processing effect of the state observer. The red wave line in Figure 12a is the measured battery impedance under experimental conditions. The impedance gradually increases as the number of cycles increases, accompanied by significant fluctuations. As the number of cycles increases, the fluctuations also become more obvious. In addition, the measurement error will also have an impact on the calculation results of impedance, which increases the difficulty of accurate estimation of the SOC. Therefore, this section will analyze the fault tolerance of the compensation factor ξ. The gray area in Figure 12a is the reasonable impedance calculated under the premise of accurate estimation of SOC, according to the DST cycle condition data under different cycle conditions of the battery. In other words, as long as the calculated result of the impedance $R$ is located in the gray area in Figure 12a, the SOC estimation value is guaranteed to be within a reasonable range. Similarly, the reasonable system compensation factor is in the gray area in Figure 12b, and the standard system compensation factor is the red wave line in Figure 12b. As the battery usage increases, the accuracy of the system model may decrease, but the observer's compensation factor ξ and impedance estimation can have a larger fault tolerance area.

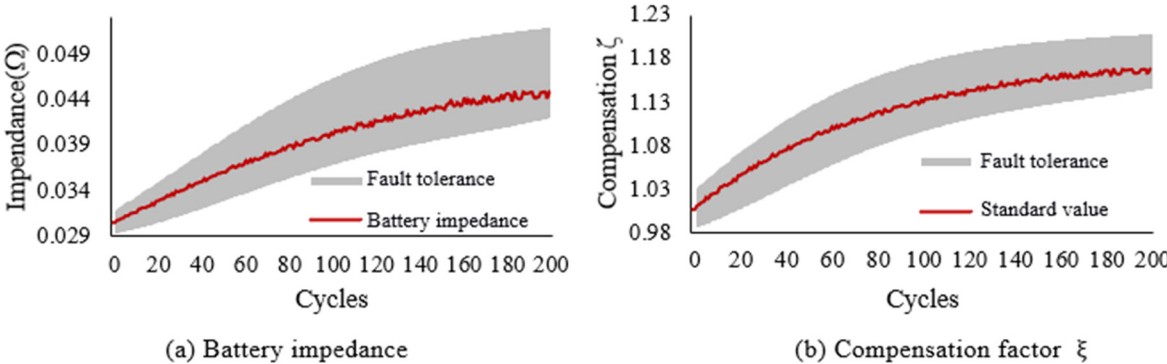

**Figure 12.** Variation characteristics and fault tolerance analysis of two-stage PI observer.

## 4. Conclusions

In order to adapt to the variability of battery state, this paper proposes a two-stage PI state observer with compensation factor to realize online estimation of battery impedance and SOC. The Dynamic Stress Test (DST) under different conditions verifies the real-time and universality of the method. The main conclusions are as follows:

1.  The experimental results show that the two-stage PI observer method can obtain reliable data results in the presence of unknown initial SOC, current drift, measurement noise, or inaccurate battery capacity.
2.  The compensation factor can adjust the model parameters online according to the battery usage, compensate part of the capacity loss and keep the system robust.
3.  The proposed SOC estimation method is capable of obtaining satisfactory accuracy in different use states for test batteries. The SOC error can be kept within 2%.
4.  The proposed SOC and battery impedance estimation have a simple structure and are easy to implement.

**Author Contributions:** Conceptualization, X.Y. and M.H.; methodology, T.C.; software, M.H.; validation, T.C., M.H. and R.W.; formal analysis, T.C.; investigation, T.C.; resources, X.Y.; data curation, R.W.; writing—original draft preparation, M.H.; writing—review and editing, T.C.; visualization, T.C.; supervision, R.W. All authors have read and agreed to the published version of the manuscript.

**Funding:** This research received no external funding.

**Institutional Review Board Statement:** Not applicable.

**Informed Consent Statement:** Not applicable.

**Data Availability Statement:** The data generated in this study cannot be shared.

**Conflicts of Interest:** The authors declare no conflict of interest.

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
