# Peer review of "A Fast Lithium-Ion Battery Impedance and SOC Estimation Method Based on Two-Stage PI Observer"

_wevj, doi:10.3390/wevj12030108_

Round 1

Reviewer 1 Report

This paper proposed a two-stage PI observer method for estimating the impedance and SOC of lithium-ion batteries. Generally, the English needs to be improved, and the following concerns need to be addressed in the revision.

  1. Further proofreading is required and the English needs to be improved. Some typos and grammatical errors can still be found throughout the manuscript. Taking only one paragraph in the Introduction for example, Line 75, batteryàbatteries; Line 77, fast the calculation speedàthe computation efficiency; Line 77, As it well knownàIt has been well known that; Line 78, affect->affects; Line 82, at->during; Line 86-87, please double check the incomplete sentence “Comparing with the method based on several observers like a sliding mode observer [20] and a dual-circuit observer [22].”.
  2. A space is needed between the word and the following reference number.
  3. When introducing the model-based battery state estimation, recent electrochemical models and relevant state estimation methods should be covered, e.g., “A physics-based distributed parameter equivalent circuit model for lithium-ion batteries” and “Constrained ensemble kalman filter for distributed electrochemical state estimation of lithium-ion batteries”. Please compare such models with the one applied in this work expressed in Equation (1).
  4. On page 4, in order to find out the relationship between the dynamic impedance and battery operating current, a set of experiments with different currents at discharge were designed. Were these circuit experiments and how were these experiments set up? Please provide more details about the testbed and testing steps for Table I as well as in Section 3.1.
  5. In the proposed method, the factor of battery aging was considered for the battery SoC estimation but the battery capacity degradation was not sufficiently reviewed. More details about the battery aging can be found, e.g., in “A fast estimation algorithm for lithium-ion battery state of health”. In the Introduction, while some SoC estimation methods for individual battery cells were reviewed, the SoC estimation for multi-battery systems considering the battery interconnection has not been discussed, such as the "Estimation of Cell SOC Evolution and System Performance in Module-Based Battery Charge Equalization Systems".
  6. The battery impedance is commonly influenced by not only the battery aging but also the battery temperature. How would the proposed method be changed if the temperature influence is considered?
  7. The proposed method still lacks a comparison with a large number of other model-based SoC estimation methods proposed in recent studies to demonstrate its advantage. Please at least provide some comparison results in terms of the estimation accuracy and computation time, etc.

Author Response

1. Further proofreading is required and the English needs to be improved. Some typos and grammatical errors can still be found throughout the manuscript. Taking only one paragraph in the Introduction for example, Line 75, battery ->batteries; Line 77, fast the calculation speed ->the computation efficiency; Line 77, As it well known ->It has been well known that; Line 78, affect->affects; Line 82, at->during; Line 86-87, please double check the incomplete sentence “Comparing with the method based on several observers like a sliding mode observer [20] and a dual-circuit observer [22].”.

ANSWER: Spelling and grammatical errors in the manuscript have been further verified

2. A space is needed between the word and the following reference number.

ANSWER: A space has been added between the word and the reference number

3. When introducing the model-based battery state estimation, recent electrochemical models and relevant state estimation methods should be covered, e.g., “A physics-based distributed parameter equivalent circuit model for lithium-ion batteries” and “Constrained ensemble kalman filter for distributed electrochemical state estimation of lithium-ion batteries”. Please compare such models with the one applied in this work expressed in Equation (1).

ANSWER: In lines 115-116 of the manuscript, the distributed parameter equivalent circuit model and the distributed electrochemical model of the lithium-ion battery have been compared with the model applied in the article

4. On page 4, in order to find out the relationship between the dynamic impedance and battery operating current, a set of experiments with different currents at discharge were designed. Were these circuit experiments and how were these experiments set up? Please provide more details about the testbed and testing steps for Table I as well as in Section 3.1.

ANSWER: This is a pulse current discharge experiment. The impedance is solved through a process of current change. Lines 228-238 of the manuscript and Figure 5(b) have explained the specific details of the experiment.

5. In the proposed method, the factor of battery aging was considered for the battery SOC estimation but the battery capacity degradation was not sufficiently reviewed. More details about the battery aging can be found, e.g., in “A fast estimation algorithm for lithium-ion battery state of health”. In the Introduction, while some SOC estimation methods for individual battery cells were reviewed, the SOC estimation for multi-battery systems considering the battery interconnection has not been discussed, such as the "Estimation of Cell SOC Evolution and System Performance in Module-Based Battery Charge Equalization Systems".

ANSWER: The article " A fast estimation algorithm for lithium-ion battery state of health " explains the exponential transformation of battery SOH with the number of cycles. It is very helpful to determine the correction coefficient in my article, which is quoted in line 208 of the article. Line 61 of the introduction also compares the battery SOC estimation in the module-based battery in the article " Estimation of Cell SOC Evolution and System Performance in Module-Based Battery Charge Equalization Systems "

6. The battery impedance is commonly influenced by not only the battery aging but also the battery temperature. How would the proposed method be changed if the temperature influence is considered?

ANSWER: The battery impedance is indeed affected by temperature. If you consider the effect of temperature, the result will be more accurate. The article can consider the temperature correction in the Arrhenius formula, and also use a correction coefficient to correct the battery impedance.

7. The proposed method still lacks a comparison with a large number of other model-based SoC estimation methods proposed in recent studies to demonstrate its advantage. Please at least provide some comparison results in terms of the estimation accuracy and computation time, etc

ANSWER: In Figure 9 of the article, the extended Kalman filter method is cited to estimate the SOC and compared with the PI observer method. Compared with the extended Kalman filter, the method proposed in the article converges to an accurate value more accurately and quickly.

Reviewer 2 Report

The paper is well written and structured, it allows for easy reading. The contents are scientifically valid and clear. However, I have some comments;  I hope the authors will consider them in order to improve their manuscript. In this sense, I ask the authors to integrate the text of the manuscript and respond to my comments one by one, leaving my question and inserting their answer immediately afterwards so that I can easily read and understand their answers.

# 1 Authors study lithium batteries, but which ones?
For example LCO (LiCoO2), LMO (LiMn2O4), NMC, LFP (LiFePO4), LTO (Li2TiO3). Does their method apply to all types or to some of them?

# 2 Check line 86, the sentence seems incomplete.

#3 Section 2. Can I resolve the two levels of PI simultaneously (at the same time) or I have to resolve them separately, in two distinct time steps?

# 4 Does the estimate of the parameters depend on the operating point of the battery? e.g. 10%, 30% 50% 90% of the rated current? In other words, does the estimate return identical results whatever the current delivered by the batteries during the test or does the estimate improve / worsen as the output current decreases / increases?
# 5 What effect does temperature have in estimating? Has the battery cooled down during the test? Is the room temperature controlled and maintained at a desired / optimal value?
# 6 How was the noise reduced in Fig. 6 and subsequent figures?
# 7 I suggest posting a photo of the lab setup
# 8 Applicability of the proposed method. The authors state in the conclusions - Line 317 - that the proposed SOC and battery impedance estimation have a simple structure and are easy to implement. What does this mean? For example, can the proposed method be implemented on a calculator of a few euros / dollars such as raspberry or Arduino? Or you need a higher performance microcontroller such as, for example, the National Instruments processors that are used to drive the ac-dc converters of 10-100kW electric motors or UPS for datacenter?

# 9 What are the performances of the proposed method when new and used batteries are considered?
Suppose I use a second life battery that has been used by an electric vehicle for three years. I now use this battery to create a residential storage system in combination with a PV system. Can I use the proposed estimator and implement it in the microcontroller of the PV & battery system?

My best regards

Author Response

# 1 Authors study lithium batteries, but which ones?
For example LCO (LiCoO2), LMO (LiMn2O4), NMC, LFP (LiFePO4), LTO (Li2TiO3). Does their method apply to all types or to some of them?

ANSWER: The battery used in the article is a ternary lithium-ion battery called ISR18650PC. At the same time, the article uses a general empirical model, this method is applicable to all types of lithium-ion batteries.

# 2 Check line 86, the sentence seems incomplete.

ANSWER: Sentence errors have been checked and corrected

#3 Section 2. Can I resolve the two levels of PI simultaneously (at the same time) or I have to resolve them separately, in two distinct time steps?

ANSWER: The article's two-level PI observers are performed simultaneously. In every second, the first-level PI observer calculates the impedance, and uses the result to estimate the battery SOC by the second-level PI observer.

# 4 Does the estimate of the parameters depend on the operating point of the battery? e.g. 10%, 30% 50% 90% of the rated current? In other words, does the estimate return identical results whatever the current delivered by the batteries during the test or does the estimate improve / worsen as the output current decreases / increases?

ANSWER: The parameter identification in the article is based on the discharge data of the resistance impedance experiment, and the least squares method is used to fit. The parameters identified by the different discharge data will have slight differences, but the general trend is the same. When estimating the SOC, use PI The observer can eliminate the influence of parameter identification errors.

# 5 What effect does temperature have in estimating? Has the battery cooled down during the test? Is the room temperature controlled and maintained at a desired / optimal value?

ANSWER: Temperature has an effect on impedance. The article does not consider the effect of temperature. If the effect of temperature is considered, a temperature correction can be considered. During the test, the battery will stand at a temperature of 25°C for one hour, and all experiments will be kept in a constant temperature environment of 25°C. So the ambient temperature of the battery is controlled at an ideal value
# 6 How was the noise reduced in Fig. 6 and subsequent figures?

ANSWER: The article adds Gaussian noise to the current data. When used to calculate impedance, the PI observer filters out the influence of Gaussian noise. The noise can be eliminated by adjusting the P value and I value of the observer.
# 7 I suggest posting a photo of the lab setup

ANSWER: Figure 4-5 has been added to the revised article, and the experimental details have been explained.

# 8 Applicability of the proposed method. The authors state in the conclusions - Line 317 - that the proposed SOC and battery impedance estimation have a simple structure and are easy to implement. What does this mean? For example, can the proposed method be implemented on a calculator of a few euros / dollars such as raspberry or Arduino? Or you need a higher performance microcontroller such as, for example, the National Instruments processors that are used to drive the ac-dc converters of 10-100kW electric motors or UPS for datacenter?

ANSWER: The PI observer is a relatively simple state estimation method. It does not require very demanding computing power. The two-stage PI observer model built using Simulink is much simpler than the extended Kalman filter, so a simple microcontroller can be implemented. . The raspberry or Arduino controller you proposed is achievable.

# 9 What are the performances of the proposed method when new and used batteries are considered?

ANSWER: When using a new battery, the impedance correction factor is 1, which does not greatly improve the battery SOC estimation. The reason is that the SOC estimation of the new battery is very accurate, but the SOC estimation of the old battery is not accurate. At this time, the correction factor The problem of inaccurate SOC estimation of old batteries is well improved. The proposed method ensures that the SOC error is kept within 2% regardless of whether it is a new or old battery.

Round 2

Reviewer 1 Report

Thanks for the response. All comments and concerns have been generally addressed by the authors. Please double check why there are two figures in Fig. 9. I have no more comments.

Author Response

1. Thanks for the response. All comments and concerns have been generally addressed by the authors. Please double check why there are two figures in Fig. 9. I have no more comments.

ANSWER: There are two figures in figure 9 , figure 9(a) and figure 9(b). Figure 9(a) shows the SOC estimation results of each method, but it is not easy to analyze their error size through figure 9(a), so I use figure 9(b) to show the error results of each method. The error size of each method can be analysed more obviously. The speed and accuracy also can be showed more obviously in figure 9(b). Thank you for your review!